# Catalysts of Conversation: Examining Interaction Dynamics Between Topic Initiators and Commentors in Alzheimer's Disease Online Communities

Submission Id: 2383

## Abstract

Informal caregivers (e.g., family members or friends) of people living with Alzheimer's Disease and Related Dementias (ADRD) face substantial challenges and often seek informational or emotional support through online communities. Understanding the factors that drive engagement within these platforms is crucial, as it can enhance their long-term value for caregivers by ensuring that these communities effectively meet their needs. This study investigated the user interaction dynamics within two large, popular ADRD communities, *TalkingPoint* and *ALZConnected*, focusing on topic initiator engagement, initial post content, and the linguistic patterns of comments at the thread level. Using analytical methods such as propensity score matching, topic modeling, and predictive modeling, we found that active topic initiator engagement drives a higher comment volume, and reciprocal replies from topic initiators encourage further commentor engagement at the community level. Practical caregiving topics prompt more re-engagement of topic initiators, while emotional support topics attract more comments from other commentors. Additionally, the linguistic complexity and emotional tone of a comment are associated with its likelihood of receiving replies from topic initiators. These findings highlight the importance of fostering active and reciprocal engagement and provide effective strategies to enhance sustainability in ADRD caregiving and broader health-related online communities.

## CCS Concepts

• **Human-centered computing** → **Social media**; • **Applied computing** → **Sociology**; • **Information systems** → *Social networking sites*.

## Keywords

Alzheimer's Disease, Online Communities, Informal Caregivers, User Engagement, Social Media Analytics

**ACM Reference Format:**
Anonymous Author(s). 2024. Catalysts of Conversation: Examining Interaction Dynamics Between Topic Initiators and Commentors in Alzheimer's Disease Online Communities. In *Proceedings of The ACM Web Conference (The Web Conference 2025)*. ACM, New York, NY, USA, 13 pages. https://doi.org/XXXXXXX.XXXXXXX

## 1 Introduction

Alzheimer's Disease and Related Dementias (ADRD) give significant challenges to both patients [12] and their families [7], generally due to the progressive nature of these conditions [13] and extensive care required [9]. This care is predominantly provided by informal caregivers, such as family members or close friends, who often report substantial emotional, physical, and financial burdens [28, 40]. As a result, there is a high demand for support and resources to help them maintain their caregiving roles and manage their own health conditions [18, 41].

Online communities have become essential platforms for informal ADRD caregivers to seek emotional and informational support [5, 35]. These communities allow caregivers to connect with numerous peer caregivers who share similar experiences and thus can provide advice on managing various caregiving tasks without the constraints of time and location [11]. Notably, reading or participating in online peer discussions can reduce caregivers' depressive symptoms [46] and improve the quality of life [3].

Given the vital role of these online communities in supporting caregivers, characterizing the factors that drive online engagement within these platforms is essential for ensuring their long-term support for caregivers [4]. Particularly, studies have shown that high messaging volumes and a strong sense of closeness gained through mutual support among informal ADRD caregivers are associated with the perceived value of these communities [23, 46]. However, few studies have focused on maintaining the sustainability of online ADRD caregiving communities. While broader research on online engagement has examined activity patterns at the community level, such as topics, user retention and community growth [10, 15, 34], these studies often lack detailed, actionable insights into how to enhance interactions at the individual level.

To fill this gap, this study focused on two large, prominent online communities for ADRD patients and their caregivers: *TalkingPoint* [2] and *ALZConnected* [1], organized by the UK Alzheimer's Society and the Alzheimer's Association, respectively. In both communities, users can initiate a topic thread with an *initial post* as a *topic initiator*, and other users can provide comments or replies as *commentors*. Topic initiators can further engage in the discussion either by replying to commentors or posting additional *self-comments* to continue the discussion. This study defines such active engagement of topic initiators as *Topic Initiator Engagement (TIE)*.

Interactions between topic initiators and commentors are crucial for maintaining active discussions. According to social presence theory [19], visible participation by key users (such as topic initiators) encourages broader community interaction. Social exchange theory [14] supports the idea that reciprocal interactions (such as receiving replies) reinforce engagement. Additionally, Pan et al. [36] found that linguistic alignment between posts and replies enhances

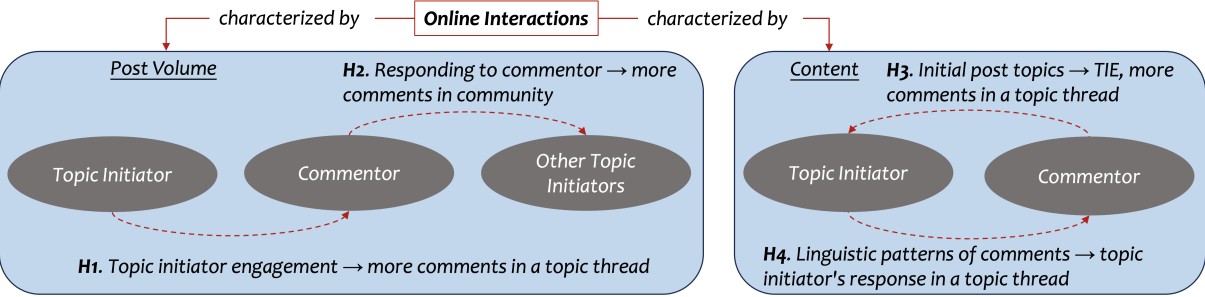

**Figure 1: An overview of the four hypotheses proposed in this study regarding interactions between topic initiators and commentors at the topic thread or community level. TIE: topic initiator engagement.**

participation in an online depression community. However, none of these theories have been examined in the context of online ADRD caregiving discussions. In this research, we apply these theories to guide the design of four hypotheses on how engagement dynamics, content, and linguistic features influence participation in ADRD caregiving communities as follows (Figure 1):

- **H1**: Threads, where topic initiators engage in following discussions, will have a higher comment volume.
- **H2**: Commentors who receive replies from a topic initiator will contribute more comments in the community.
- **H3**: Topics of initial posts are associated with both TIE and the comment volume of topic threads.
- **H4**: Linguistic patterns of comments are associated with the likelihood of obtaining replies from topic initiators.

By identifying the factors that encourage communications between topic initiators and commentors, this study aims to provide actionable strategies to enhance online engagement and collective knowledge sharing, particularly in supporting informal ADRD caregivers. The findings may extend beyond ADRD communities and provide broader implications for enhancing online user interactions and community sustainability in various health-related domains.

**Ethical and Privacy Consideration**. This study was exempt from human subjects research by the Institutional Review Board of [Institution Name]. All quotes were rephrased to ensure anonymity, and experiments were conducted on secure servers.

## 2 Data Source

To ensure the generalizability of our findings, we use data from two popular online ADRD communities: *TalkingPoint* and *ALZConnected*, to conduct experiments. These platforms provide valuable resources for caregivers and individuals discussing ADRD in several countries, including the United States, United Kingdom and Canada. They offer a rich dataset for analyzing engagement patterns. We focused solely on self-identified caregivers and excluded administrators, moderators, and users who merely published advertisements. We collected publicly accessible data using a web crawler built with the `BeautifulSoup` v4.11 Python package in 2022.

The *TalkingPoint* dataset contains 846,344 posts, including 81,068 *initial posts* and 765,276 comments, published by 34,551 unique users between March 31, 2003 and November 3, 2022. Of these users, 27,907 initiated at least one *topic thread*, and 26,651 participated

as commentors. A total of 53,997 threads had TIE, while 27,071 did not. While the *ALZConnected* dataset includes 521,382 posts, with 56,928 *initial posts* and 464,454 comments, published by 18,586 unique users between November 14, 2011 and August 6, 2022. Of these users, 12,634 were topic initiators, while 14,897 participated as commentors. 30,696 threads had TIE, while 26,232 did not.

## 3 Topic Initiator Engagement (H1)

### 3.1 Data Cohort

In this analysis, we compared topic threads with TIE to non-TIE threads in terms of their association with comment volume. The ratio of TIE to non-TIE threads was approximately 2:1 in *TalkingPoint* (53,997 vs. 27,071) and 1.2:1 in *ALZConnected* (30,696 vs. 26,232).

To control for potential confounding factors influencing comment volume, we generated four key features: (1) *Topic Proportions*, (2) *Sentiment Scores*, (3) *Previous Activity*, and (4) *Initial Post Length*. For *Topic Proportions*, we applied Latent Dirichlet Allocation (LDA) to initial posts and used perplexity scores to determine the optimal number of topics, ultimately selecting 11 topics from a range of [5, 40] to minimize perplexity and achieve a clear topic structure [49]. We removed the least frequent topic from the analysis due to the fact that the sum of all the topic probabilities equals one. *Sentiment Scores* were conducted using Valence Aware Dictionary and sEntiment Reasoner (VADER) [20], producing positive (Pos.), negative (Neg.) and compound (Comp.) sentiment scores to capture the emotional tone of each post. *Previous Activity* was measured by calculating the number of comments made by the topic initiator before starting the thread, serving as a proxy for user reputation and accounting for the possibility that highly active users may naturally attract more comments. *Initial Post Length* was also considered because longer posts might attract more responses. These features were used to control for external factors, all features were normalized before performing PSM.

### 3.2 Propensity Score Matching

We applied PSM to control for confounding variables and isolate the effect of TIE on comment volume. PSM is a widely recognized technique for addressing selection bias by creating comparable groups [8]. We adopted Logistic Regression (LR) to estimate propensity scores based on the selected covariates. After matching with searching for the nearest neighbor (NN) in the non-TIE group per TIE, we

achieved a TIE to non-TIE ratio of 53,997 to 19,486 for *TalkingPoint* and 30,696 to 14,825 for *ALZConnected*.

To assess covariate balance after matching, we calculated the Standardized Mean Difference (SMD) for each covariate. It has been suggested that SMDs smaller than 0.20–0.25 generally indicate a good matching [42]. All features had SMD values below 0.2 for both platforms, indicating a good balance of features between the matched groups [45]. Detailed SMD values for each feature were shown in Appendix A.1.

### 3.3 Statistical Analysis

We performed a Wilcoxon rank sum test to evaluate the difference in the number of comments (excluding comments made by topic initiators themselves) between topic threads with and without TIE. For *TalkingPoint*, the average number of comments in threads with TIE was 8.9 (median: 6.0, IQR: 3.0–10.0), while in non-TIE threads was 3.8 (median: 3.0, IQR: 1.0–5.0). The Wilcoxon rank sum test resulted in a p-value < 0.001, indicating a significant difference between the two groups. Similarly, in *ALZConnected*, the average number of comments in threads with TIE was 8.9 (median: 6.0, IQR: 4.0–10.0), compared to 4.6 (median: 3.0, IQR: 2.0–6.0) in non-TIE threads. P-value < 0.001, also indicating a significant difference between the two groups.

These results confirm **H1** in both communities, showing that the engagement of topic initiators in their own threads is associated with increasing comment volume. This analysis highlights the importance of TIE in encouraging community interactions and sustaining discussion threads.

## 4 Commentor Engagement (H2)

### 4.1 Data Cohorts

*4.1.1 Inferring Reply Relationships.* We constructed a dataset to map reply relationships within the community discussions. This dataset facilitated the understanding of interaction dynamics by identifying who replied to whom and which comments were responded to by topic initiators. In *TalkingPoint*, we found that 31% of the total comments (181,069 out of 584,207) were replied to by a topic initiator, involving 29% of the commentors (4,902 out of 16,889). Similarly, in *ALZConnected*, we identified that 21% of comments (79,095 out of 385,359) were replied to by topic initiators, involving 19% of commentors (2,242 out of 11,580).

*4.1.2 Time Window Setup and Non-TIE Simulation.* We divided the timeline of each commentor's posting activity into defined time windows relative to the date of the **first** TIE. Time windows, ranging from 1 month to 6 months (or 30 to 180 days in 30-day increments), were defined to capture the periods "before" ($P_{before}$) and "after" ($P_{after}$) receiving the TIE.

For the non-TIE group, we simulated a comparable "pseudo-TIE date", generated based on the distribution of days between the first composed comment and the first TIE date for users in the TIE group. This ensures that non-TIE group users were observed under similar conditions, facilitating the creation of a fair comparison.

*4.1.3 Confounding Factors.* To control for potential confounding factors influencing changes in future online engagement, we constructed several features to represent commentors' behavior in

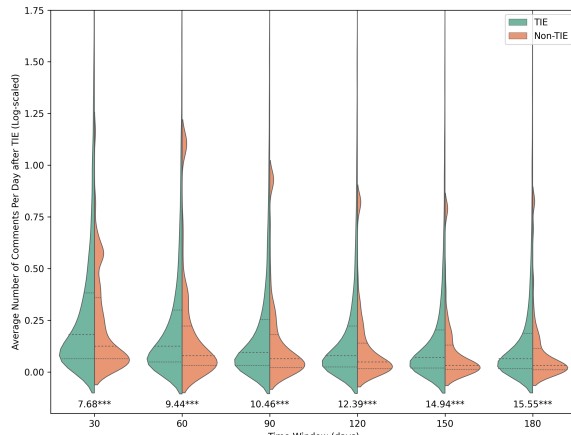

**Figure 2: The log-scaled average number of comments per day after receiving TIEs across time windows in *TalkingPoint* ($^{***}p < 0.001$, $^{**}p < 0.01$, $^{*}p < 0.05$).**

$P_{before}$. These features included: (1)*Topic Proportions*, which has 15 topics from a range of [5, 40] generated using an LDA model optimized with perplexity measurements (similar to in **H1**), excluding the lowest proportion topic for PSM to avoid multicollinearity, (2)*Sentiment Scores* using VADER (Pos., Neg., and Comp.), (3)*Comments Length* (number of words per comment), and (4)*Absolute Time*, which measures the number of months since the platform's creation up to the posting time, capturing the potential influence of time on overall community activities. Additionally, we computed (5)Freq. $P_{b}efore$, the average frequency of comments posted per time window, and (6)Median Absolute Deviation (MAD) to capture variability in posting behavior during $P_{before}$. All features were normalized before performing PSM to find the nearest neighbors.

### 4.2 Propensity Score Matching

To evaluate how receiving a TIE influences commentors' future engagement, which is measured as the number of comments published during $P_{after}$, we applied PSM, as introduced in **H1**, to compare commentors who received TIEs with those who did not (the non-TIE group). Again, for each commentor, we aggregated the mean values of features (1)–(4) and combined them with (5) and (6) to represent overall behavior for matching. As presented in Appendix A.3, the majority of SMD values fall below 0.2, reflecting a strong balance of features between the matched groups across time windows and platforms. We used LR to compute propensity scores based on the constructed features and applied NN to find the nearest commentor from the non-TIE group.

### 4.3 Statistical Analysis

Figure 2 shows the log-scaled average number of comments per day after receiving TIEs $P_{after}$ in *TalkingPoint*, with the TIE group consistently exhibiting higher engagement than the Non-TIE group across all time windows. We used a log-scaled visualization to better highlight the differences between the groups while mitigating the impact of outliers, which could obscure the visibility of the

trends in the raw values. For example, at the 30-day window in *TalkingPoint*, the mean number of comments per day for the TIE group is 0.45 (median: 0.20, *IQR*: 0.07–0.47), while the Non-TIE group has a mean of 0.30 (median: 0.13, *IQR*: 0.07–0.43). At the 180-day window, the mean for the TIE group is 0.25 (median: 0.07, *IQR*: 0.02–0.21), whereas the Non-TIE group has a mean of 0.12 (median: 0.03, *IQR*: 0.01–0.12). Similarly, for *ALZConnected*, at the 180-day window, the TIE group's mean is 0.35 (median: 0.09, *IQR*: 0.03–0.33), compared to the Non-TIE group with a mean of 0.20 (median: 0.05, *IQR*: 0.01–0.21). Appendix A.2 presents a similar analysis for *ALZConnected*, where the long-term engagement pattern for TIE and Non-TIE groups also follows a similar trend.

Across all time windows, Wilcoxon rank-sum tests reveal statistically significant differences between the two groups' comment frequencies, consistently with $p < 0.001$. These results strongly support **H2**, confirming that receiving replies from topic initiators is associated with a higher average number of comments per day, demonstrating a substantial increase in engagement compared to the Non-TIE group.

## 5 Topic Analysis of Initial Posts (H3)

### 5.1 Data Cohort

We focused on initial posts from *TalkingPoint* and *ALZConnected*. For each topic thread, we collected the content of the initial post, whether the thread had TIE, and the total number of comments composed by other commentors. The number of comments and the presence of TIE served as metadata, forming the foundation for analyzing how topics and content characteristics of initial posts impact the online engagement of topic initiators and commentors. In *TalkingPoint*, there is an average of 7 comments per thread, with a median of 4.0 (*IQR* 2.0–8.0); In *ALZConnected*, there is an average of 7 comments per thread, with a median of 5.0 (*IQR* 2.0–8.0).

### 5.2 Structural Topic Modeling

Structural Topic Model (STM) is a method applied to uncover latent topics within text data, allowing for the inclusion of document-level metadata to explore how topics interact with other variables [38]. It is widely adopted in computational social media research [22, 33, 37]. STM is particularly useful in **H3** as it enables analysis of not only the content but also how the presence of TIE or comment volume interacts with uncovered topics.

*5.2.1 Model Training.* We trained our STM models using the stm v1.3.7 package in R, with the initialization method set to LDA. The use of LDA as the initialization method enables flexible topic discovery while ensuring a stable starting point. To determine the optimal number of topics $k$ in stm model, we applied the method of selecting proper topic sizes [32] from a range of [5, 40] using two key metrics: 1) *exclusivity* and 2) *semantic coherence*. Exclusivity measures how uniquely the top words belong to a specific topic, with higher scores indicating more distinct topics. Semantic coherence assesses how often the top words of a topic appear together in the same documents, with higher coherence suggesting that the words are meaningfully related. These metrics were selected to provide a better balance between distinct topic generation and interpretability compared to traditional LDA evaluation metrics like

perplexity (which focused on statistical fit rather than the clarity of topics). Our analysis indicated that $k = 30$ provided the best balance, capturing diverse topics while maintaining interpretability.

*5.2.2 Generated Topics.* The STM model identified 30 distinct topics within the *TalkingPoint* community. Due to the large number of topics, we grouped them into five main categories for clarity: *Community Engagement*, *Practical Care*, *Feelings*, *Health Discussions*, and *Legal & Financial Matters*. An *Others* category was created for topics that primarily consist of linguistic common words like "*don't*" and "*cannot*", which do not carry clear interpretative meanings. Our analysis focused mainly on the five primary categories. Our categorization process was a collaboration with sociology experts specializing in community engagement and healthcare communication. We jointly reviewed the top 20 words and posts for each topic, forming an initial set of categories. These categories were then refined through three rounds of expert feedback to ensure alignment with the core themes of the dataset and to accurately capture the nuances of the topics. Figure 3 visualizes the distribution of these topics. A similar set of topics was generated in *ALZConnected*, as shown in Appendix A.4, indicating that ADRD communities share common themes, reinforcing the generalizability of these findings.

The analysis revealed that community engagement plays a key role in these discussions. Topic 19, labeled *Community Words*, captures 6.5% of the overall discussion. This demonstrates the platform's role in encouraging interaction, as users often discuss the community itself and its value for connecting with others. In addition to community topics, *Practical Care* emerged as the most dominant category, accounting for 23.5% of the overall discussions. Within this category, Topic 27 (*Caregiver Support*) makes up 4.5% of the conversations, and Topic 25 (*Safety Concerns*) represents 2.7%. The prevalence of these topics shows that many users seek advice and shared experiences on daily caregiving tasks, safety concerns and support, underlining the platform's practical value. The emotional side of caregiving is another significant theme, particularly in the *Feelings* category. Topic 16 (*Family Feelings*) accounts for 4.3% of the discussions. A typical post reads, "*Hi everyone, I just wanted to let everyone know how I feel right now ... my heart feels like it's breaking...(rephrased).*" This category captures the emotional struggles of caregivers, as they share feelings and receive support, highlighting the community's role in containing emotional as well as practical help. *Health Discussions* focus on medical treatments and health management, with Topic 29 (*Treatment Options*) representing 3.7% of discussions. Posts often address dementia progression and medical interventions. One user, for example, writes, "*[Drug-name] inhibitors (e.g., [brandname]): These drugs work by increasing the level of a chemical in the brain in people with vascular dementia (rephrased).* " This demonstrates the platform's usefulness for exchanging crucial health information and advice on treatments. Finally, *Legal & Financial Matters* are discussed less frequently but still represent a significant portion of the conversation. Topics like Topic 4 (*Legal Matters*; 3.5%) and Topic 10 (*Financial Support*; 3.4%) reflect users seeking advice on issues beyond caregiving. Users often look for professional guidance to manage legal and financial concerns related to their caregiving responsibilities.

Overall, these findings suggest that ADRD communities serve a **dual** role: they are not only a source of practical caregiving advice

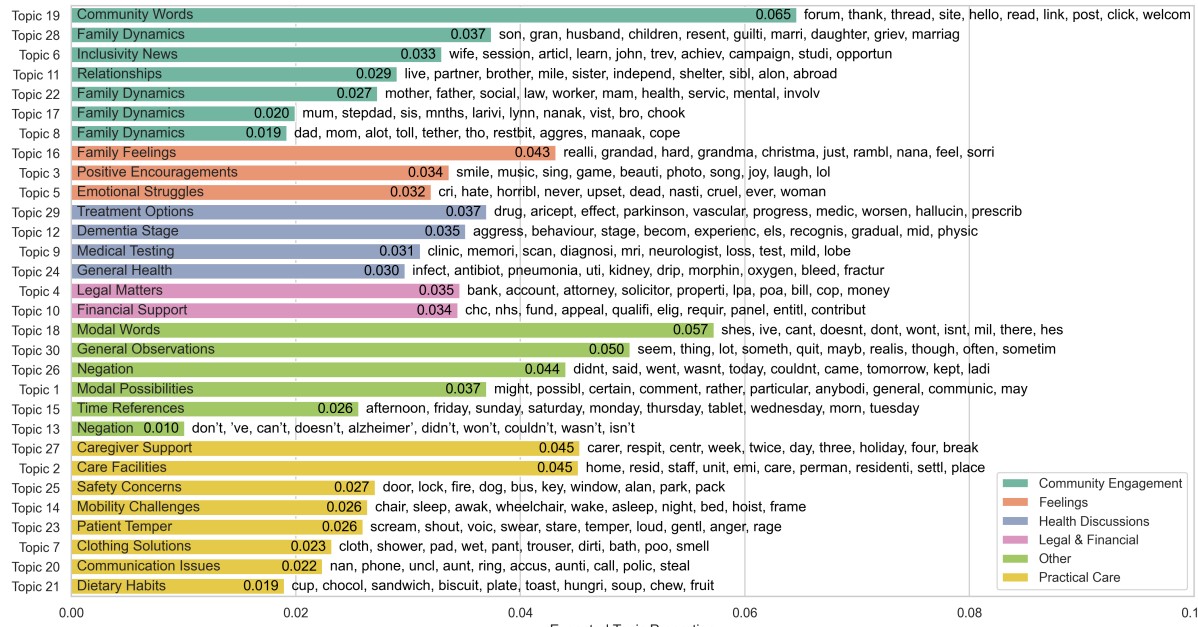

**Figure 3: Expected topic proportions for the 30 topics identified in *TalkingPoint*. Each bar represents the proportion of documents assigned to a given topic, with the top words listed on the right and color-coded by six topic categories. The axis labels the topic numbers, followed by brief summaries.**

but also a space for emotional support. Whether addressing community engagement, practical care, emotional challenges, or legal and financial concerns, the discussions reflect the comprehensive support these platforms provide to caregivers.

## 5.3 Topic Trends

*5.3.1 TIE and Topics.* Figure 4 presents the estimated impact of TIE across various topics. We used `estimateEffect` [39], which computes the difference in topic proportions between groups defined by the covariate (TIE/non-TIE), We set the plot method to `differences` to assess the effect of document-level binary covariate TIE on topic proportions. To highlight the more notable topics, we do not show the *Others* category.

Topic 24 (*General Health*) has the strongest positive correlation with TIE, with an estimate of 0.004 and $p < 0.001$. An example post reads, "*My [age] year old [Relative] is still in hospital and according to the nurse he is doing really badly ... Does anyone else have a family problem like this? (rephrased)*". Users often seek advice on health management, which drives engagement. Conversely, Topic 19 (*Community Words*) shows the strongest negative correlation, with an estimate of -0.011 and $p < 0.001$. Discussions about general community matters tend to receive less follow-up from the initiator.

In the positive significant topics, the *Practical Care* category is well-represented. Almost all topics in this category, except Topic 2 (*Care Facilities*) ($p >= 0.05$), show a positive correlation with TIE. This trend implies that practical caregiving discussions, such as those about safety concerns or caregiver support, are more likely to prompt the topic initiator to re-engage, possibly due to the direct and actionable nature of the content discussed.

On the other hand, several topics within the *Community Engagement* category, including Topic 3 (*Positive Encouragements*), Topic 6 (*Inclusivity News*), and Topic 28 (*Community Words*), are negatively correlated with TIE. For example, Topic 6 includes posts such as "*I attended the second meeting of the newly formed '[Location] Leaders Group', a group that hopes to become the voice of people with dementia in [Location] and ultimately the voice of people with dementia in [Location] (rephrased).*" These types of announcements and broader community updates typically do not require or receive further engagement from the topic initiator.

The *Feelings* category also includes two negatively correlated topics: Topic 16 (*Family Feelings*) and Topic 3 (*Emotional Struggles*). Although sharing emotions might be expected to encourage interaction, TIE measures post-initial engagement, and users may avoid further interaction due to emotional exhaustion or the nature of the emotions expressed.

Appendix A.5 contains a similar analysis in *ALZConnected*, where comparable positive and negative topic relationships with TIE are observed, demonstrating the consistency and generalizability of these findings across different online ADRD communities.

*5.3.2 TIE and Number of Comments Trend.* Figure 5 illustrates the trend in comment proportions for topics within the *Feelings* and *Health Discussions* categories. Using `continuous` in `estimateEffect`, we examined the relationship between topic proportions and the continuous variable (number of other comments).

In Figure 5 (left), all topics within the *Feelings* category, such as Topic 16 (*Family Feelings*) and Topic 3 (*Positive Encouragements*), display a consistent increase in their proportion as the number of

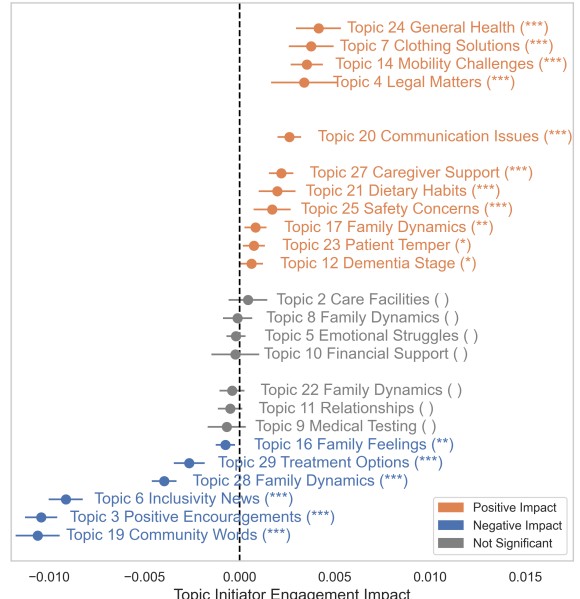

**Figure 4: The impact of Topic Initiator Engagement (TIE) across different topics. The plot shows the estimated effect of TIE on various topics ($^{***}p < 0.001$, $^{**}p < 0.01$, $^{*}p < 0.05$).**

comments grows. This is particularly notable because, as indicated in the previous section, the *Feelings* category was associated with lower levels of TIE. Nonetheless, the upward trend in the number of comments suggests that these topics may evoke strong empathy in the community, potentially encouraging members to participate and share their experiences even if the topic initiator no longer continues to participate. This observation demonstrates the compassion and support of this community, as users feel compelled to contribute to these discussions, likely to provide comfort and emotional support to others.

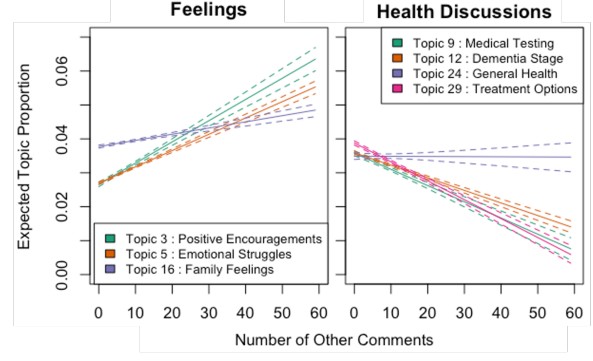

**Figure 5: Trends in comment proportions for topics within the *Feelings* and *Health Discussions* in *TalkingPoint*.**

By contrast, Figure 5 (right) shows the comment trends for topics within the *Health Discussions* category. This category exhibits a clear downward trend in comment engagement as the discussion

progresses, including Topic 9 (*Medical Testing*), Topic 12 (*Dementia Stage*), Topic 24 (*General Health*) and Topic 29 (*Treatment Options*), show decreasing engagement over time. One possible explanation for this trend is that, while these topics initially attract attention due to their informative nature, they may tend to be discussed for shorter periods as users quickly obtain the necessary information and move on. This shows that health-related discussions, although critical, do not sustain long-term engagement, as users may be more focused on acquiring specific knowledge rather than continuing extended dialogues.

These findings reveal that topics with the most comments do not always encourage TIE. For instance, while *Feelings* topics generate substantial community interaction, they do not typically lead to ongoing engagement from the topic initiator, likely because they involve emotional sharing rather than sustained discussion. On the other hand, *Health Discussions*, although critical, tend to meet users' informational needs quickly, resulting in shorter engagement. In *ALZConnected*, similar trends were observed (see Appendix A.6), with subtle differences in the proportionate increase in *Feelings* topics and the engagement length for *Health Discussions*. These differences reflect variations in community composition and discussion dynamics across platforms.

Overall, these trends highlight the complexity of participation in online ADRD communities. Some topics may not encourage follow-ups from the topic initiator but still generate meaningful community interactions, reflecting the varied roles users play in these discussions.

## 6 Linguistic Analysis of Comments (H4)

### 6.1 Data Cohort

We selected threads in which the topic initiator did reply to some comments but not all. This approach allows us to ensure that only threads in which the initiator is active (with TIE) are included, thereby controlling for the possibility that some topic initiators may simply choose not to engage further regardless of comment content. We focused on the comments made before the first explicit reply from the topic initiator in each thread. Specifically, we grouped all comments that did not receive the topic initiator's reply into a "negative" group, while the first comment that received a comment from the topic initiator serves as the "positive" case. This process enables for a comparison between the comments that receive a reply from the topic initiator against those that do not. In *TalkingPoint*, this process resulted in 23,374 pairs of negative comment group and positive case. Similarly, in *ALZConnected*, we obtained 11,868 pairs.

### 6.2 Feature Selection

To evaluate the linguistic features of comments, we focus on a set of key metrics that capture a range of aspects of language use. These include (1) emotional tone [29], (2)syntactic structure [6], (3) sentiment, (4) readability [26], and (5) linguistic similarity to the initial post. We used NRCLex [30] to categorize the emotional content of each comment into ten primary emotions: fear, anger, anticipation, trust, surprise, positive, negative, sadness, disgust, and joy. This helped assess the emotional tone of the comments. Part-of-speech

tags were generated using NLTK to identify nouns, verbs, adjectives, and adverbs, providing insight into how language complexity may influence engagement. Sentiment analysis was performed with VADER, which provided scores for positive, negative, and compound sentiment. Readability was evaluated using the Flesch Reading Ease (FRE) and Gunning Fog Index (GFI), which measure how easy or difficult the text is to read. Finally, we calculated linguistic similarity between comments and the initial post using cosine similarity based on Term Frequency-Inverse Document Frequency vectorization, providing a measure of how closely the comments aligned with the initial post's content.

For each thread, the features of negative cases were averaged to create a single composite feature vector to match the same size as positive cases.

## 6.3 Model Training

We built machine learning models to predict whether a comment would receive a reply from the topic initiator. The selected models include Logistic Regression (LR), Support Vector Classifier (SVC), Decision Tree, Gradient Boosting, AdaBoost, Extra Trees, Random Fores, and XGBoost.To ensure the robustness and validity of our results, we applied stratified shuffle to split the dataset into ten pairs of training and testing sets with a ratio of 80 to 20. Hyperparameter tuning for each model was performed using five-fold cross-validation on the training dataset. Table 1 presents the model performance of Area Under the Receiver Operating Characteristic Curve (AUC) for both *TalkingPoint* and *ALZConnected*. The XGBoost model emerged as the best-performing model, achieving the highest AUC score of 0.85 in the *TalkingPoint* dataset and 0.87 in the *ALZConnected* dataset. The optimal parameters for XGBoost included a learning rate of 0.1, a maximum depth of 3, and 200 estimators. The consistency of these results across both communities demonstrates the robustness of the XGBoost model and its ability to generalize effectively across different datasets. We subsequently applied XGBoost to the entire dataset to interpret the model and identify the most important features contributing to the prediction.

**Table 1: Model performance summary of AUC for *Talking-Point* and *ALZConnected*.**

| Model | TalkingPoint | ALZConnected |
|---|---|---|
| XGBoost | 0.850 ±0.026 | 0.867 ±0.014 |
| Gradient Boosting | 0.847 ±0.027 | 0.867 ±0.021 |
| Random Forest | 0.842 ±0.025 | 0.858 ±0.014 |
| Extra Trees | 0.837 ±0.024 | 0.851 ±0.021 |
| AdaBoost | 0.830 ±0.021 | 0.848 ±0.016 |
| Decision Tree | 0.793 ±0.026 | 0.795 ±0.023 |
| LR | 0.593 ±0.018 | 0.601 ±0.012 |
| SVC | 0.552 ±0.017 | 0.567 ±0.011 |

## 6.4 Model Interpretation with SHAP

To gain insights into the factors driving our model's predictions, we applied SHapley Additive exPlanations (SHAP) [25], a game-theoretic approach designed to explain both individual and global

predictions made by machine learning models. Figure 6 shows the SHAP summary plot for the *TalkingPoint* community, ranking features by their importance. The x-axis represents SHAP values, where red (blue) dots indicate high (low) feature values.

*Reading Ease (FRE)* is the most influential feature, with higher scores (indicating more complex comments) associated with a greater likelihood of receiving replies. Similarly, *Reading Grade (GFI)* also shows that more difficult-to-read comments are more likely to get a reply. These findings suggest that comments with higher linguistic complexity tend to engage the topic initiator more effectively; The *NRC* emotional categories, such as *Surprise (NRC)*, *Disgust (NRC)*, and *Joy (NRC)*, show predominantly blue dots on the negative SHAP values, indicating that less emotional comments are likely to receive replies. This aligns with the *Compound Sentiment (VADER)* sentiment score, where higher scores positively influence the likelihood of receiving a reply. These results suggest that comments with lower emotional content, whether positive or negative, are less engaging for the topic initiator; The part-of-speech (PoS) features, such as *Noun Percentage (PoS)* and *Verb Percentage (PoS)*, have a minimal impact on the model's predictions. This indicates that the syntactic structure of the comments is less relevant to whether they receive a reply from the topic initiator.

Interestingly, *Cosine Similarity*, still shows some influence, albeit weaker than readability and sentiment features. Higher cosine similarity values are associated with negative SHAP values, indicating that comments with content more similar to the initial post are less likely to receive replies. By contrast, posts with low similarity are more likely to attract replies from topic initiators.

These findings are consistent with those observed in the *ALZConnected* community, as detailed in Appendix A.7, allowing us to confidently conclude that readability and emotional content are more critical factors in predicting whether a comment will receive a reply, rather than its linguistic similarity to the initial post.

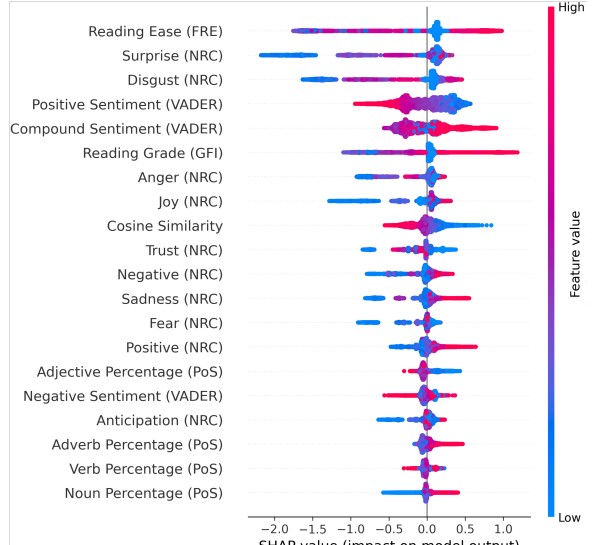

**Figure 6: SHAP values of feature importance in predicting the likelihood of comment replies.**

# 7 Discussion and Conclusions

## 7.1 Primary Findings

This study investigated the dynamics of user engagement in two large, popular online ADRD communities, *TalkingPoint* and *ALZ-Connected*, by examining four primary hypotheses. Our findings present a comprehensive understanding of how various factors, including topic initiator engagement, the content of initial posts, and the linguistic characteristics of comments, influence interaction levels within these communities.

Regarding *Topic Initiator Engagement* (**H1**), our analysis showed that threads where topic initiators participated in the following discussions accumulated significantly more comments compared with those where topic initiators did not engage in the following discussions. This result aligns with social presence theory [19], which suggests that visible, active participation by key individuals (in our case, topic initiators) within a community can enhance their presence and thus encourage the participation of others. In this context, the topic initiator's engagement can be seen as a catalyst for increased community activities, highlighting the need to encourage initiators to be involved in their own threads.

For *Commentor Engagement* (**H2**), our findings show that when commentors received replies from topic initiators, their overall commenting activities within a community increased. This finding emphasizes the importance of encouraging reciprocal interactions in sustaining user engagement over time. It is also supported by social exchange theory [14], which proposes that positive reinforcement, such as receiving a reply, motivates further participation within a social context. Moreover, reciprocal interactions are found to be correlated with positive health-related behavior in an online breast cancer forum [48]. This highlights the need for community management strategies that not only encourage posting but also promote positive reciprocal engagement to maintain an energetic community environment.

In examining the *Content of Initial Posts* (**H3**), our findings show that the initial post topics significantly impact subsequent commenting behaviors. Topics related to *practical caregiving, health discussions, and emotional support* are commonly discussed within threads. Caregivers may share their feelings as a form of release but are less likely to continue these emotional discussions by replying to others. Conversely, emotional support topics, while less likely to encourage further engagement from topic initiators, tend to attract more comments from other users. These findings also demonstrate the capability of online peer discussions for caregivers to seek informational or emotional support.

In the analysis of *Linguistic Features of Comments* (**H4**), we found that more complex and emotional comments are more likely to receive a reply from the topic initiator, while comments that are linguistically similar to the initial post are less likely to receive such a reply. While these findings are against communication accommodation theories [36], it can be explained that in online health communities, online users such as informal ADRD caregivers may pay more attention to the comments that contain more effective informational and emotional support. Such rich content makes these comments unlikely to be similar to initial posts in both content and linguistics. This may differentiate online health communities from

other online communities, such as politics, sports or music, where people focus more on discussing similar topics.

These findings show that topic initiators and commentators can stimulate each other's online activities through active engagement, and particularly, the topic and linguistic choices in their posts.

## 7.2 Related Works

*7.2.1 Online ADRD Caregiving Discussions.* Prior studies have mainly focused on content analyses of online ADRD caregiving discussions, with the majority focusing on learning the emotional and informational needs of informal ADRD caregivers and how online peer support can meet these needs [24, 44, 47]. Also, there are studies comparing discussions between different online ADRD caregiving communities [11, 31], different ADRD caregiver kin relationships [32], and caregivers and non-caregivers [16].

While these studies provided valuable insights into **what** informal ADRD caregivers discussed, they generally neglected **how** online caregiving communities can support active engagement over time. The dynamics that contribute to maintaining a vibrant and supportive community are as critical as understanding the topics being discussed. Our study moves beyond content analysis by considering the factors that are related to a sustainable online community, offering a more comprehensive understanding of how they can remain active and supportive over time.

*7.2.2 Online Engagement Dynamics.* Research on sustaining online communities has mainly focused on factors that influence user retention and participation, with an emphasis on aspects like user loyalty, shared interests, the perceived richness of online discussions, and community structures [21, 27, 43, 50].

However, these studies generally examine online engagement at the *community level*, without delving into the specific dynamics of individual discussions. Our study diverges from this perspective by focusing on the *thread level*, which represents the smallest organizational unit within an online community. We uncovered how discussions between topic initiators and commentors in a topic thread can be encouraged, which generates rich online collective knowledge to support informal ADRD caregivers.

## 7.3 Limitations and Future Works

While this study provides valuable insights, there are several avenues for further exploration. First, the analysis is based on data from two specific ADRD communities, which may limit the generalizability to other-purpose platforms. Future research could examine if the interactive patterns hold in different contexts. Additionally, factors such as demographic data were not considered. Including these in future observational studies could offer a more complete understanding of TIE dynamics.

In the meanwhile, the insights gained from the linguistic analysis of comments could contribute to the development of automated community management systems. Future research could also explore AI-driven interventions aimed at fostering community engagement. For instance, algorithms could be designed to identify posts requiring additional attention, ensuring that users receive adequate attention and support in these crucial discussions.

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

## A Appendices

### A.1 Standardized Mean Difference (SMD) for Covariates in H1

Propensity score matching (PSM) was employed in this study for **H1** to balance the covariates between the TIE and non-TIE groups, ensuring that any observed differences in engagement are attributable to the engagement itself rather than underlying differences between the groups. One method to assess the effectiveness of PSM is through the standardized mean difference (SMD), which measures the difference in covariates between the two groups on a standardized scale. SMD values below 0.20–0.25 [42] are generally considered indicative of good covariate balance, meaning that the matched groups are comparable in terms of their baseline characteristics.

Table 2 presents the SMD values for the key covariates in both *TalkingPoint* and *ALZConnected* after matching. The results show that all covariates have SMD values well below 0.05 in both communities, indicating that the matching process was successful in achieving covariate balance.

**Table 2: Standardized Mean Differences (SMD) for covariates in *TalkingPoint* and *ALZConnected*. Note that the least frequent topic (Topic #02 in TalkingPoint and Topic #04 in ALZConnected) was excluded from PSM.**

| Covariate | TalkingPoint | ALZConnected |
|---|---|---|
| *Previous Activity* | 0.013 | 0.004 |
| *Initial Post Length* | 0.001 | 0.019 |
| Sentiment Comp. | 0.000 | 0.010 |
| Sentiment Pos. | 0.014 | 0.014 |
| Sentiment Neg. | 0.007 | 0.008 |
| Topic #01 | 0.001 | 0.008 |
| Topic #02 | - | 0.002 |
| Topic #03 | 0.011 | 0.002 |
| Topic #04 | 0.008 | - |
| Topic #05 | 0.009 | 0.023 |
| Topic #06 | 0.003 | 0.006 |
| Topic #07 | 0.000 | 0.001 |
| Topic #08 | 0.003 | 0.012 |
| Topic #09 | 0.013 | 0.001 |
| Topic #10 | 0.004 | 0.001 |
| Topic #11 | 0.007 | 0.020 |

### A.2 Commentor Engagement in *ALZConnected*

Figure 7 presents the log-scaled average number of comments per day after receiving the first TIE, comparing the TIE and Non-TIE groups across different time windows. While some shorter time windows, such as the 20-day window, show no significant difference between the two groups (mean of 0.67 for the TIE group and 1.20

for the Non-TIE group, with a negative statistic -2.15), the pattern becomes more aligned with the results observed in *TalkingPoint* as the time window extends. For instance, by the 100-day window, the TIE group demonstrates a higher mean of 0.42 (median: 0.13, *IQR*: 0.04–0.42) compared to the Non-TIE group, which drops to a mean of 0.32 (median: 0.08, *IQR*: 0.02–0.33). This suggests that while some short-term windows do not show a significant difference, the overall trend in longer time windows indicates sustained engagement for the TIE group, similar to the pattern seen in *TalkingPoint*.

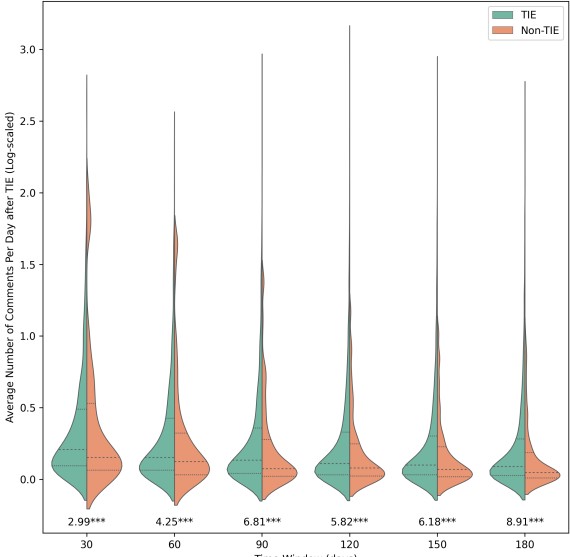

**Figure 7: Log-scaled average number of comments per day after TIE in *ALZConnected*. Green represents the TIE group, and orange represents the Non-TIE group. Wilcoxon rank-sum statistics are provided below each window, with significance levels indicated (***$p < 0.001$, **$p < 0.01$, *$p < 0.05$).**

### A.3 Standardized Mean Difference (SMD) for Covariates in H2

Table 3 shows the Standardized Mean Differences (SMD) for various covariates across different Time Windows for both the *TalkingPoint* and *ALZConnected* platforms. The $Window = X$ refers to the specific time window (e.g., X=30, 60, 90, etc.), indicating the number of days or periods being analyzed. Features like *Sentiment Pos.* and *Sentiment Neg.* represent the positive and negative sentiment of the posts, while *Abs. Time* captures the absolute time since the community creation. *MAD* stands for Mean Absolute Deviation, reflecting the variability in the number of posts, and Freq. *$P\_before$* refers to the frequency of comments posted before a reference time point. Since the topic proportions must sum to 1 across all topics, including all topics could introduce redundancy and potential multicollinearity in the model, as each topic's proportion is inherently dependent on the others. To avoid this issue, we removed the lowest proportion topic from each community, as it contributed minimally to the overall topic distribution and would not provide significant additional variance for analysis. Specifically, *Topic #5* was removed

from *TalkingPoint* and *Topic #14* from *ALZConnected*. This ensures that the remaining topic proportions provide meaningful variance without introducing unnecessary dependencies in the model, thus enhancing the robustness of the feature set.

In general, the SMD values across most covariates are below 0.2, which indicates strong balance between the groups. For instance, features such as *Comments Length* and *Sentiment Comp.* consistently show low SMD values across all windows, demonstrating a stable match between groups. Similarly, *MAD* values mostly stay below the threshold, although in the *ALZConnected* platform, *MAD* for $Window = 60$ is slightly larger than 0.2 (0.217), which suggests a modest imbalance in post variability during this time period.

A few other features demonstrate relatively larger imbalances, but still all below 0.25 [17]. For example, in the *TalkingPoint* platform, *Topic #14* shows an elevated SMD value in the $Window = 60$ (0.182), and *Topic #03* also displays higher differences in longer windows, such as $Window = 150$ (0.172) and $Window = 180$ (0.242). On the *ALZConnected* platform, *Topic #01* exhibits slightly higher SMDs, especially in the longer time windows like $Window = 180$, where it reaches 0.216.

Despite these larger imbalances in a few features, the majority of covariates exhibit low SMD values, reinforcing the overall balance and comparability of the matched groups. This balance ensures the robustness of the results across different time windows.

## A.4 Generated Topics in *ALZConnected*

The STM generated 30 distinct topics for the *ALZConnected* community, which are visually represented in Figure 8. The distribution of these topics closely mirrors the patterns observed in the *TalkingPoint* community, highlighting the consistency of caregiver discussions across different platforms.

Each topic reflects a unique theme within the broader discourse of Alzheimer's Disease and Related Dementias (ADRD). These themes range from practical caregiving advice, emotional support, and health discussions to legal and financial concerns, all of which are crucial areas of interest for caregivers navigating the challenges of ADRD.

The bar chart illustrates the proportion of discussions dedicated to each topic, providing insights into the dominant areas of focus in the *ALZConnected* community. For example, topics related to practical care, such as *Community Engagement* and *Feelings*, continue to be prevalent, similar to what we observed in *TalkingPoint*. Likewise, Health Discussions topics, including *Diagnosis Progress* and *Patient Symptoms*, account for a significant portion of the community's interactions.

## A.5 TIE with Topic Relationships in *ALZConnected*

Figure 9 presents a detailed visualization of the impact of Topic Initiator Engagement (TIE) on various topics within the *ALZConnected* community.

The topics with positive impacts (orange) show a strong association with increased TIE. Practical caregiving discussions, such as *Drug Dosage*, *Hospital Surgery*, and *Cleaning Challenges*, are highly correlated with topic initiator re-engagement. In contrast, the topics with negative impacts (blue) show that certain emotionally driven

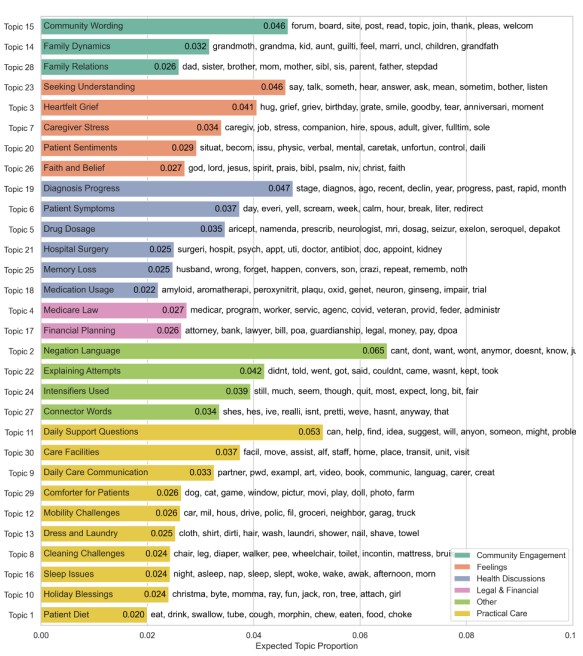

**Figure 8: Distribution of expected topic proportions for the 30 identified topics in the *ALZConnected* community.**

or community-focused topics tend to result in less engagement from topic initiators. Discussions such as *Faith and Belief*, *Heartfelt Grief*, and *Community Wording* exhibit lower levels of re-engagement.

These pattern of engagement is consistent with what we observed in the *TalkingPoint* community, reinforcing the conclusion that practical caregiving discussions are more likely to associate with ongoing re-engagement.

## A.6 TIE with Number of Comments in *ALZConnected*

Figure 10 displays the trends in the proportion of comments for topics within the *Feelings* and *Health Discussions* categories. This figure offers a comparative view of how these discussions evolve as the number of comments increases. Unlike the clear distinctions observed in the *TalkingPoint* community, the trends in *ALZConnected* are more muted, particularly within the *Feelings* category, suggesting differences in community dynamics between these platforms.

In the *Feelings* category (upper panel), topics such as *Seeking Understanding* and *Heartfelt Grief* show a gradual increase in engagement as the number of comments grows. However, the upward trend is less pronounced compared to *TalkingPoint*, indicating that while these topics resonate with community members, they may not provoke as sustained a response. Similarly, in the *Health Discussions* category (lower panel), topics like *Drug Dosage*, *Hospital Surgery*, and *Diagnosis Progress* show steady trends as comment volume increases. Although these health-related discussions are crucial to the community, their engagement pattern in *ALZConnected* appears less dynamic than in *TalkingPoint*, where such topics see more significant fluctuations over time.

**Table 3: Standardized Mean Differences (SMD) across Time Windows for TalkingPoint and ALZConnected. Note that the least frequent topic (Topic #05 in TalkingPoint and Topic #14 in ALZConnected) was excluded from PSM.**

| Feature | $Window = 30$ | $Window = 60$ | $Window = 90$ | $Window = 120$ | $Window = 150$ | $Window = 180$ |
|---|---|---|---|---|---|---|
| *TalkingPoint* | | | | | | |
| *Comments Length* | 0.122 | 0.077 | 0.064 | 0.062 | 0.115 | 0.075 |
| Sentiment Pos. | 0.043 | 0.052 | 0.059 | 0.065 | 0.073 | 0.033 |
| Sentiment Neg. | 0.023 | 0.085 | 0.119 | 0.112 | 0.127 | 0.175 |
| Sentiment Comp. | 0.043 | 0.021 | 0.047 | 0.069 | 0.068 | 0.103 |
| Abs. Time | 0.054 | 0.026 | 0.022 | 0.064 | 0.025 | 0.064 |
| Topic #01 | 0.100 | 0.128 | 0.126 | 0.131 | 0.165 | 0.188 |
| Topic #02 | 0.148 | 0.001 | 0.026 | 0.000 | 0.030 | 0.050 |
| Topic #03 | 0.066 | 0.090 | 0.149 | 0.170 | 0.172 | 0.242 |
| Topic #04 | 0.037 | 0.061 | 0.077 | 0.096 | 0.011 | 0.033 |
| Topic #06 | 0.022 | 0.005 | 0.035 | 0.038 | 0.016 | 0.046 |
| Topic #07 | 0.066 | 0.048 | 0.034 | 0.012 | 0.049 | 0.051 |
| Topic #08 | 0.083 | 0.073 | 0.057 | 0.065 | 0.058 | 0.061 |
| Topic #09 | 0.024 | 0.011 | 0.015 | 0.023 | 0.025 | 0.047 |
| Topic #10 | 0.065 | 0.023 | 0.002 | 0.071 | 0.051 | 0.087 |
| Topic #11 | 0.034 | 0.043 | 0.025 | 0.003 | 0.012 | 0.011 |
| Topic #12 | 0.027 | 0.032 | 0.019 | 0.010 | 0.059 | 0.049 |
| Topic #13 | 0.031 | 0.052 | 0.042 | 0.108 | 0.051 | 0.105 |
| Topic #14 | 0.064 | 0.182 | 0.153 | 0.092 | 0.095 | 0.041 |
| Topic #15 | 0.035 | 0.024 | 0.029 | 0.003 | 0.001 | 0.025 |
| Freq. $P_{before}$ | 0.103 | 0.057 | 0.061 | 0.065 | 0.069 | 0.090 |
| MAD | 0.107 | 0.095 | 0.066 | 0.051 | 0.047 | 0.043 |
| *ALZConnected* | | | | | | |
| *Comments Length* | 0.043 | 0.008 | 0.013 | 0.103 | 0.023 | 0.072 |
| Sentiment Pos. | 0.102 | 0.112 | 0.055 | 0.068 | 0.005 | 0.021 |
| Sentiment Neg. | 0.070 | 0.070 | 0.109 | 0.096 | 0.128 | 0.165 |
| Sentiment Comp. | 0.042 | 0.092 | 0.082 | 0.049 | 0.082 | 0.058 |
| Abs. Time | 0.026 | 0.110 | 0.071 | 0.068 | 0.130 | 0.128 |
| Topic #01 | 0.091 | 0.120 | 0.128 | 0.180 | 0.177 | 0.216 |
| Topic #02 | 0.037 | 0.007 | 0.059 | 0.048 | 0.018 | 0.012 |
| Topic #03 | 0.002 | 0.023 | 0.034 | 0.057 | 0.064 | 0.037 |
| Topic #04 | 0.004 | 0.011 | 0.023 | 0.054 | 0.032 | 0.022 |
| Topic #05 | 0.062 | 0.054 | 0.097 | 0.064 | 0.098 | 0.111 |
| Topic #06 | 0.037 | 0.064 | 0.048 | 0.004 | 0.044 | 0.024 |
| Topic #07 | 0.143 | 0.048 | 0.060 | 0.011 | 0.001 | 0.003 |
| Topic #08 | 0.102 | 0.059 | 0.031 | 0.041 | 0.020 | 0.038 |
| Topic #09 | 0.041 | 0.053 | 0.058 | 0.090 | 0.066 | 0.067 |
| Topic #10 | 0.022 | 0.031 | 0.023 | 0.034 | 0.013 | 0.005 |
| Topic #11 | 0.187 | 0.154 | 0.114 | 0.083 | 0.105 | 0.143 |
| Topic #12 | 0.079 | 0.053 | 0.042 | 0.097 | 0.053 | 0.056 |
| Topic #13 | 0.098 | 0.089 | 0.106 | 0.142 | 0.123 | 0.164 |
| Topic #15 | 0.032 | 0.048 | 0.034 | 0.019 | 0.003 | 0.042 |
| Freq. $P_{before}$ | 0.139 | 0.084 | 0.066 | 0.054 | 0.048 | 0.033 |
| MAD | 0.066 | 0.217 | 0.023 | 0.188 | 0.079 | 0.126 |

## A.7 SHAP Summary in *ALZConnected*

Figure 11 presents the SHAP summary plot for the *ALZConnected*. The plot reveals similar trends to those observed in *TalkingPoint*, with linguistic complexity and emotional tone being crucial factors in predicting engagement.

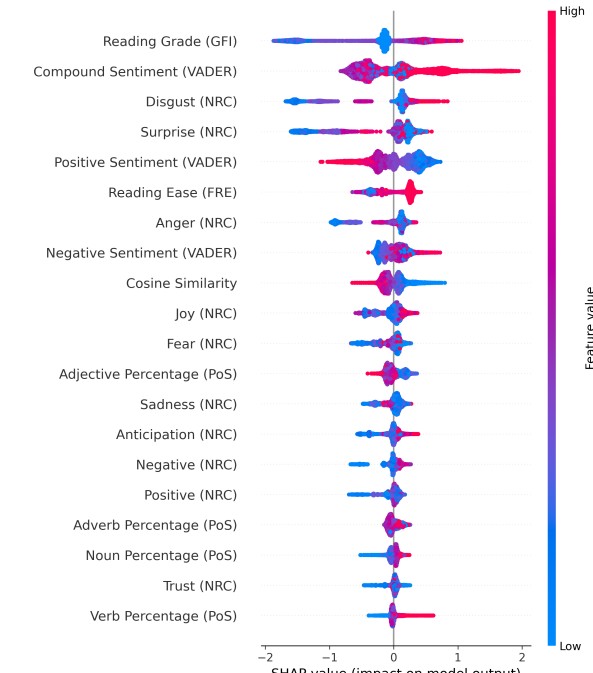

**Figure 11: Analysis of SHAP values to determine the feature importance in predicting the likelihood of receiving a reply within *ALZConnected*.**

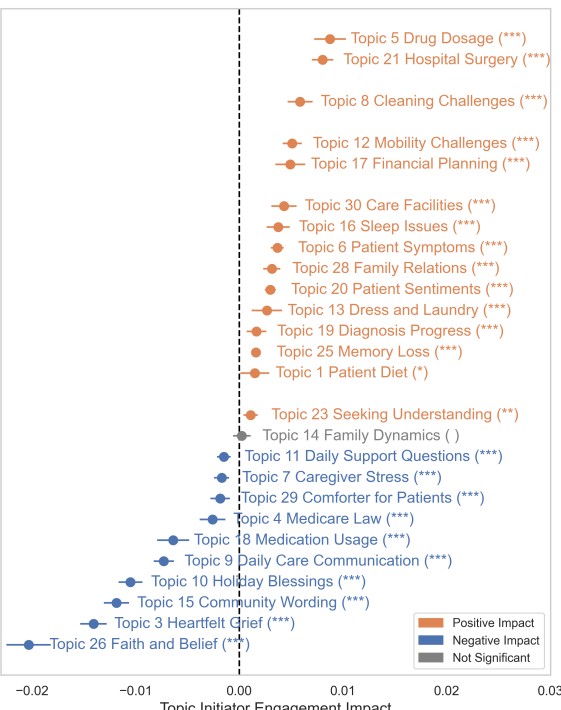

**Figure 9: Visualization of the TIE impact on various topics within *ALZConnected*. The plot shows each topic's sensitivity to TIE ($^{***}p < 0.001$, $^{**}p < 0.01$, $^{*}p < 0.05$).**

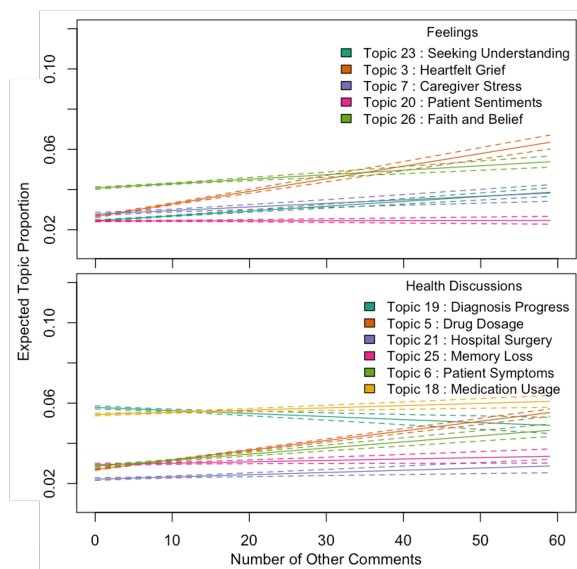

**Figure 10: Comparative trends of comment frequencies over time in the *Feelings* and *Health Discussions* categories within *ALZConnected*. Each line represents a topic, with trends showing how discussion frequency changes with the number of other comments in the forum.**

