# OpenReview forum: "Catalysts of Conversation: Examining Interaction Dynamics Between Topic Initiators and Commentors in Alzheimer's Disease Online Communities"
_ACM.org/TheWebConf/2025/Conference — WWW 2025 Poster_

### Official Review · Reviewer_gmrb · 2024-11-14

**Novelty:** 3
**Technical Quality:** 5

**Review:**

This paper explores the factors affecting user engagement activity in online communities for Alzheimer's Disease and Related Dementias (ADRD). The authors identify four potential influencing factors: whether the topic initiator engages in subsequent discussions, whether commentors receive replies from the topic initiator, the topics of initial posts, and the linguistic characteristics of comments. They employ machine learning algorithms such as LDA and LR to analyze the collected data and discuss the impact of these factors on post popularity. Overall, the writing is smooth, the structure is clear, and it is easy to understand. However, the paper focuses primarily on the analysis of ADRD forum data, resulting in a narrow research scope and limited practical applicability of the conclusions. Although the experiments and analyses are relatively rigorous, there is a lack of technical innovation, as the algorithms used, such as LDA and LR, are quite traditional.

**Questions:**

1. The paper investigates the factors influencing user engagement activity in online communities for Alzheimer's Disease and Related Dementias (ADRD). However, since ADRD is a relatively niche condition, is the scope of this research too narrow? Additionally, what impact does this engagement activity have on the users?
2. Please explain the practical application value of the four hypotheses proposed in the paper and their corresponding conclusions, as well as any improvement suggestions that can be derived from them.
3. In line 148, it is stated that the study aims to propose actionable strategies to enhance online engagement. Please clarify what actionable strategies can be derived from your research conclusions.
4. The technical methods used in the paper are somewhat outdated. For instance, in Section 6.3, LR, SVC, and XGBoost are employed to predict whether a comment would receive a reply from the topic initiator. Why not utilize more advanced deep learning methods, such as pre-trained models like BERT or large language models (LLMs)?
5. Please emphasize the innovation and uniqueness of your research methods.
6. When using existing tools such as NLTK, FRE, GFI, and TF-IDF in Section 6.2, please include appropriate references.
7. In line 752, SHAP is regarded as a metric to evaluate the likelihood of comment replies. Please briefly explain the mechanism of SHAP and its specific role in this experiment. Additionally, it is necessary to clarify the meaning of SHAP values, particularly the distinction between positive and negative values and what a high SHAP value signifies.
8. Section 7 "Discussion and Conclusion" is too lengthy. Please separate the experimental findings and conclusions into two distinct sections: one independent "Conclusion" section to succinctly summarize the paper, and another section to discuss the research results.

**Reviewer Confidence:**

3: The reviewer is confident but not certain that the evaluation is correct

**Scope:**

2: The connection to the Web is incidental, e.g., use of Web data or API

---

### Official Review · Reviewer_EXmt · 2024-11-20

**Novelty:** 3
**Technical Quality:** 4

**Review:**

**Summary**

The paper analyze the interactions occurring in two different online platforms targeting ADRD (Alzheimer’s Disease and Related Dementias) communities, that are TalkingPoint and ALZConnected.

The paper focuses on the causes and effects of Topic Initiator Engagement (TIE). The topic initiator is the user who writes the post (initial post). A topic initiator triggers a TIE if she further engages in the discussion under her post.

The study targets four hypotheses, about possible patterns associated to TIE, such as increased comment volume (H1), increased users’ activity (H2). topical features (H3) and other linguistic patterns (H4).

**Strengths:**

The paper is generally well written, with a clear language and accurate organization.

All the results are properly presented and supported. Each insight of the paper is accompanied by a precise description of the methodology used.

The paper adopts interesting dataset, and contributes in giving more knowledge about the niche of online interactions of ADRD communities with well-established methodologies and broad data.

The results are also related to sociological theories, such as social presence theory and social exchange, providing an evident connection between social science and social media data analysis.

**Weaknesses:**

The paper does not propose any novel methodologies, and relies on widely used techniques, such as, Propensity Score Matching, Latent Dirichlet Allocation, Sentiment Analysis (VADER), Structural Topic Model, XGBoost. I underline that Propensity Score Matching is not universally accepted [1].

Some of these methods have also been improved in the literature, and more advanced techniques could also surpass their performance. For this reason, the work offers a data analysis of the two online communities, while not posing significant advancements from a methodological point of view.

H1 compares the length of the threads between posts with a TIE and a control group. However, it is never explicited how the authors deal with the fact that posts with TIE have at least a comment by definition (while this is not controlled in the propensity score matching). The results from section 3 seems to be obvious, since posts with a TIE should have more comments because of TIE itself.

**Comments:**

A deeper presentation of TalkingPoint and ALZConnected would be useful for readers that are not familiar with these platforms. What is the structure of the threads? Are comments directed only to the posts or could they direct other comments? This information is necessary for understanding the engagements.

From Figure 3 I see that Topic 19 contains words such as “site”, “link”, “click”. Is it possible that these words are related to common URLs that are linked inside the platform? I suggest to rename the Topics, since it is not trivial to find the Topic from the plot. I could suggest to name them in the same order they are plotted, or divided by cathegory (Community_Engagement.1, Community_Engagement.2,…, Feelings.1, Feelings.2,…).

I would appreciate some more statistics or plots about collected data (in section 2 or in the appendix), like the distribution of thread length, or the time series of the volume of posts in the period 2003 - 2022, or some user-level statistics.

**Minor comments:**

In the text and in the caption the authors refer to Figure 2 as “log-scaled”, but the axes are in linear scale. Moreover, all the violinplots consistently present a bimodal distribution in the non-TIE plot. What is the reason behind the upper smaller peak? The labels, ticklabels and legend of Figure 2 are not easy to read, I suggest to increase the fontsize to make it readable.

In section 6 the authors “selected threads in which the topic initiator did not reply to some comments but not all” is not clear. Is this due to the fact that topic initiator comments to the specific comments? Until this moments I was thinking that the conversations were linear.

In 7.2.2 the authors could provide more extensive comparison with existing non-ADRD literature. Note that the “engagement” is often referred as “interaction” in the literature. For example, in Reddit studies, researchers often consider an “interaction” a comment directed to a post, and many papers examines the threads’ dynamics.

**Scope**

- *4: The work is relevant to the Web and to the track, and is of broad interest to the community*

- *3: The work is somewhat relevant to the Web and to the track, and is of narrow interest to a sub-community*

- *2: The connection to the Web is incidental, e.g., use of Web data or API*

- *1: The work is irrelevant to the Web*


2: the contribution of the paper is mainly related to the dynamics of engagement in ADRD communities.

**Novelty**
*Please assign a numerical rating from 1 (worst) to 7 (best) to indicate the novelty of the contributions this paper makes to the research area being studied, i.e., asking an important question, a significant originality of ideas and/or execution.*

3: in the interest of WWW community, the main novelty refers to the definition of TIE and its connection with community engagement.

**Technical Quality**
*Please assign a numerical rating from 1 (worst) to 7 (best) to indicate the soundness of the technical claims, experimental and research methodology and whether the main claims of the paper are sufficiently supported with evidence.*

4: the results are well-presented and the methods used are correct and properly reported. Some robustness analysis would give more strength to the results and the plots could be improved.

**Reviewer Confidence**
*Please indicate how confident you are in your evaluation.*

- *4: The reviewer is certain that the evaluation is correct and very familiar with the relevant literature*

- *3: The reviewer is confident but not certain that the evaluation is correct*

- *2: The reviewer is willing to defend the evaluation, but it is likely that the reviewer did not understand parts of the paper*

- *1: The reviewer's evaluation is an educated guess*


3: the reviewer is familiar with the literature concerning dynamics of interactions in social media.

[1] King, Gary, and Richard Nielsen. "Why propensity scores should not be used for matching." *Political analysis* 27.4 (2019): 435-454.

**Questions:**

A deeper presentation of TalkingPoint and ALZConnected would be useful for readers that are not familiar with these platforms. What is the structure of the threads? Are comments directed only to the posts or could they direct other comments? This information is necessary for understanding the engagements.

From Figure 3 I see that Topic 19 contains words such as “site”, “link”, “click”. Is it possible that these words are related to common URLs that are linked inside the platform? I suggest to rename the Topics, since it is not trivial to find the Topic from the plot. I could suggest to name them in the same order they are plotted, or divided by cathegory (Community_Engagement.1, Community_Engagement.2,…, Feelings.1, Feelings.2,…).

I would appreciate some more statistics or plots about collected data (in section 2 or in the appendix), like the distribution of thread length, or the time series of the volume of posts in the period 2003 - 2022, or some user-level statistics.

**Reviewer Confidence:**

3: The reviewer is confident but not certain that the evaluation is correct

**Scope:**

2: The connection to the Web is incidental, e.g., use of Web data or API

---

### Official Review · Reviewer_5L7n · 2024-11-24

**Novelty:** 5
**Technical Quality:** 3

**Review:**

This paper introduces the dynamic interaction between topic initiators and commentators in Alzheimer's disease online communities, with novel and unique ideas that effectively enhance sustainability in ADRD caregiving and broader health-related online communities. The paper has a clear and reasonable organizational structure, and the writing is clear.

Advantage:
1) The paper has a novel and unique idea, and by analyzing the dynamic interaction between the topic initiators and commentators, it is expected to help informal caregivers.
2) The paper proposes four hypotheses based on social presence theory and social exchange theory to examine how participation dynamics, content, and language features affect informal caregivers in ADRD caring communities.
3) The paper is detailed and rich in content.

Disadvantages:
1) How are the four hypotheses related to the social presence theory and social exchange theory? Why these four hypotheses?
2) These findings can only guide communities on how to intervene, but fail to guide online individuals on how to have a certain impact on informal caregivers in ADRD caring communities.
3) What is the practical significance of informal caregivers by these findings?

**Questions:**

1) How are the four hypotheses related to the social presence theory and social exchange theory? Why these four hypotheses?
2) These findings can only guide communities on how to intervene, but fail to guide online individuals on how to have a certain impact on informal caregivers in ADRD caring communities.
3) What is the practical significance of informal caregivers by these findings?

**Reviewer Confidence:**

2: The reviewer is willing to defend the evaluation, but it is likely that the reviewer did not understand parts of the paper

**Scope:**

4: The work is relevant to the Web and to the track, and is of broad interest to the community

---

### Official Review · Reviewer_vjvS · 2024-11-28

**Novelty:** 6
**Technical Quality:** 6

**Review:**

Summary  This study investigates interaction dynamics within two online Alzheimer's disease caregiving communities, TalkingPoint and ALZConnected. The research evaluates four hypotheses, focusing on the effects of topic initiator engagement (TIE), reciprocal interactions, thematic content, and linguistic features on community participation. Using methods such as propensity score matching and, topic modeling, the study provides actionable insights for enhancing caregiver engagement. Findings include that TIE and practical caregiving topics significantly increase engagement, while linguistic complexity and emotional content of comments influence responses. The work is positioned as a resource for sustaining community dynamics in health-related online platforms.

Positives

The research is timely and addresses the critical need for sustaining online support communities for informal caregivers of Alzheimer’s patients. Insights have implications for improving user retention and interaction quality across health-related online communities (and probably beyond health communities).

The use of propensity score matching (PSM) to control for confounding factors strengthens the validity of findings. studies like this need to be carefully design to account for confounders and I am glad the authors take good care with a good design.

The findings are actionable, with clear recommendations for fostering engagement, such as prioritizing reciprocal interactions and supporting practical caregiving topics.

Negatives

I do not have many negatives except some minor ones.

Emotional topics are identified as less conducive to TIE, but the analysis does not delve deeply into why or how this could be addressed. For example, emotional exhaustion or stigma in sharing feelings could be better contextualized.

Similarlyl, the finding that linguistic complexity drives replies from topic initiators is intriguing but warrants further exploration. For example, does complexity signal credibility, or does it align with higher emotional stakes?

**Questions:**

Emotional topics are identified as less conducive to TIE, but the analysis does not delve deeply into why or how this could be addressed. For example, emotional exhaustion or stigma in sharing feelings could be better contextualized.

Similarlyl, the finding that linguistic complexity drives replies from topic initiators is intriguing but warrants further exploration. For example, does complexity signal credibility, or does it align with higher emotional stakes?

**Reviewer Confidence:**

3: The reviewer is confident but not certain that the evaluation is correct

**Scope:**

3: The work is somewhat relevant to the Web and to the track, and is of narrow interest to a sub-community

---

### Official Review · Reviewer_PnWG · 2024-12-02

**Novelty:** 4
**Technical Quality:** 4

**Review:**

This study examines user engagement dynamics in two ADRD caregiver communities, TalkingPoint and ALZConnected. It finds that active involvement by topic initiators increases comment volume, and reciprocal replies encourage further engagement. Practical caregiving topics prompt more re-engagement from initiators, while emotional support topics attract more comments from others. Linguistic complexity and emotional tone in comments also influence the likelihood of receiving replies from initiators. The study proposes four hypotheses on how engagement, content, and linguistic features affect participation. These findings offer strategies to enhance community sustainability, with broader implications for health-related online platforms.
Here are a few comments on this paper:

1. This paper primarily relies on human-extracted features for analysis, which may overlook important contextual information present in the raw text. Additionally, this paper primarily utilizes existing methods and tools, which may limit the introduction of novel approaches or innovative solutions.

2. In Section 3, the description of the four key features—Topic Proportions, Sentiment Scores, Previous Activity, and Initial Post Length—is clear. However, it would strengthen the explanation to provide a brief rationale for the selection of each feature. Additionally, in the subsequent discussion of the methods used to calculate each feature, it would be helpful to include a justification for choosing these specific methods over other possible alternatives.

3. In Section 5, the categorization of the 30 topics into five broad categories (Community Engagement, Practical Care, Feelings, Health Discussions, and Legal & Financial Matters) provides useful insights. However, the distinction between categories like "Health Discussions" and "Practical Care" could be more explicitly defined. For example, the boundary between caregiver support (Practical Care) and treatment options (Health Discussions) might need clearer justification.

**Questions:**

1. In Section 4, the description of the pseudo-TIE date is somewhat vague. This step would benefit from further clarification regarding the simulation process. For example, is it assumed that the TIE and non-TIE groups exhibit similar posting behaviors prior to receiving a TIE?
2. Are there notable differences in user engagement dynamics between TalkingPoint and ALZConnected that may be linked to platform design, community guidelines, or user demographics? What can these differences tell us about community sustainability in online health forums?
3. Emotional support is essential for AD patients. Can we expect emotional support topics to lead to long-term community growth, even if they don't directly lead to engagement from topic initiators? What role do emotional threads play in creating a sense of community belonging? Any findings on that?

**Reviewer Confidence:**

4: The reviewer is certain that the evaluation is correct and very familiar with the relevant literature

**Scope:**

4: The work is relevant to the Web and to the track, and is of broad interest to the community